# Effects of Face Masks on Physical Performance and Physiological Response during a Submaximal Bicycle Ergometer Test

**DOI:** 10.3390/ijerph19031063

**Published:** 2022-01-18

**Authors:** Benjamin Steinhilber, Robert Seibt, Julia Gabriel, Joulia Brountsou, Markus Muljono, Tomasz Downar, Mona Bär, Rosina Bonsch, Adrian Brandt, Peter Martus, Monika A. Rieger

**Affiliations:** 1Medical Faculty, Institute of Occupational and Social Medicine and Health Services Research, University Hospital Tuebingen, 72074 Tuebingen, Germany; robert.seibt@med.uni-tuebingen.de (R.S.); Julia.gabriel@med.uni-tuebingen.de (J.G.); joulia.brountsou@student.uni-tuebingen.de (J.B.); markus.muljono@med.uni-tuebingen.de (M.M.); tomasz.downar@med.uni-tuebingen.de (T.D.); mona.baer@med.uni-tuebingen.de (M.B.); rosina.bonsch@student.uni-tuebingen.de (R.B.); adrian.brandt@t-online.de (A.B.); monika.rieger@med.uni-tuebingen.de (M.A.R.); 2Medical Faculty, Institute for Clinical Epidemiology and Applied Biometry, University Hospital Tuebingen, 72076 Tuebingen, Germany; peter.martus@med.uni-tuebingen.de

**Keywords:** physical working capacity, COVID-19, occupational health and safety, personal protective equipment

## Abstract

The ongoing COVID-19 pandemic requires wearing face masks in many areas of our daily life; hence, the potential side effects of mask use are discussed. Therefore, the present study explores whether wearing a medical face mask (MedMask) affects physical working capacity (PWC). Secondary, the influence of a filtering facepiece mask with exhalation valve class 2 (FFP2exhal) and a cotton fabric mask (community mask) on PWC was also investigated. Furthermore, corresponding physiological and subjective responses when wearing face masks as well as a potential moderating role of subjects’ individual cardiorespiratory fitness and sex on face mask effects were analyzed. Thirty-nine subjects (20 males, 19 females) with different cardiorespiratory fitness levels participated in a standardized submaximal bicycle ergometer protocol using either a MedMask, FFP2exhal, community mask, or no mask (control) on four days, in randomized order. PWC130 and PWC150 as the mechanical load at the heart rates of 130 and 150 beats per minute were measured as well as transcutaneous carbon dioxide partial pressure, saturation of peripheral capillary oxygen, breathing frequency, blood pressure, perceived respiratory effort, and physical exhaustion. Using the MedMask did not lead to changes in PWC or physiological response compared to control. Neither appeared changes exceeding normal ranges when the FFP2exhal or community mask was worn. Perceived respiratory effort was up to one point higher (zero-to-ten Likert scale) when using face masks (*p* < 0.05) compared to control. Sex and cardiorespiratory fitness were not factors influencing the effects of the masks. The results of the present study provide reason to believe that wearing face masks for infection prevention during the COVID-19 pandemic does not pose relevant additional physical demands on the user although some more respiratory effort is required.

## 1. Introduction

After the onset of the COVID-19 pandemic, politicians and medical experts established rules to avoid the spreading of the coronavirus (SARS-CoV-2) in order to protect the population [1,2,3]. In this context, face masks have become an important measure since they are effective in reducing respiratory transmission by droplet infection and aerosols [4], and in many areas of life, such as work, wearing face masks has become mandatory [2]. Current recommendations for infection prevention at the workplace in Germany either suggest wearing a medical face mask (MedMask), also called surgical mask, or filtering facepieces class 2 (FFP2) or N95 masks to protect oneself or others, respectively [5]. Besides the protective effects of the face masks, undesirable side effects are discussed [6], as the masks have a direct effect on breathing air supply. The scientific evidence on the effects of wearing face masks on worker health, work ability, and performance is controversial. Two recent reviews, one scoping review focusing on negative effects by face masks and a systematic review analyzing the effects of face masks during exercise, have summarized the current scientific evidence on this topic with different perspectives. While Sha, et al. [7] reported that masks have only small effects on physiological responses and no effect on performance during exercise, Kisielinski et al. [8] concluded that face masks can “have a negative effect on the basis of all aerobic life (…) with physical, psychological, and social consequences for the individual human being” [8] (p. 35). 

Despite these contradictive views, wearing face masks is generally recommended [2,9]. With respect to the workplace setting, the Coordination Center for Biological Hazards of the German Statutory Accident Insurance (DGUV) recommends that the masks used for general infection protection during the pandemic are worn no longer than 2 h at a time. Then, a 30-min period without mask is required before the mask can be worn for another period of 2 h. Thus, a total of three wearing periods per working day can take place [10]. This pandemic-specific recommendation was based on another document by the German Statutory Accident Insurance addressing occupational health and safety when wearing respirators [11]. In this document, wear-time limitations are suggested for the different classes of respirators to prevent overloading of the users. Hence, the wear-time limits suggested in the latter document are based on working conditions where FFP2 masks with exhalation valve (FFPexhal) are used as protective measures against hazardous dust, which simply has been adopted as the wear-time recommendation for any face mask during the COVID-19 pandemic. This recommendation, which has not been supported by scientific evidence so far, significantly affects the daily work in German companies from different occupational sectors.

As seen for many of the governmental regulations during the pandemic, recommendations may change when more scientific data become available. A major problem of the currently available scientific literature dealing with side effects of face masks is a substantial risk of bias, as shown by Shaw, Zello, Butcher, Ko, Bertrand, and Chilibeck [7]. This risk refers to small sample sizes (insufficient statistical power); low methodological quality; testing the masks under maximum performance, which is not representative for occupational tasks; using masks that are currently not recommended during the COVID-19 pandemic [12]; or improper use of the masks resulting from the applied scientific methods, e.g., wearing a respiratory mask over the face mask, leading to unrealistic leakage [13,14]. Despite the mentioned shortcoming of the available literature and the scoping review by Kisilinsky et al. (2021) focusing on negative outcomes related to face masks, the majority of available data indicate minor physiological responses within a non-relevant range by using face masks [7,15]. Even less response has been associated by using a MedMask compared to FFP2 or N95 masks [7,16].

In addition, an interaction of wearing face masks with individual factors, such as a person’s cardiorespiratory fitness level, age, or sex, known to modify physical capacity [17,18,19,20,21,22] has almost not been considered when evaluating face masks [15,16]. The controversial scientific debate about face masks and their impact on employees’ health [23] indicates the need for further high-quality studies as a solid basis for proper recommendations during the COVID-19 pandemic. Therefore, this study investigated whether face masks would impair physical performance and affect physiological and subjective response during submaximal physical activity as well as a potential moderating role of cardiorespiratory fitness level and sex on the mask effects.

## 2. Materials and Methods

### 2.1. Participants

Thirty-nine healthy subjects (20 men; 19 women) were included. All gave their written informed consent prior to participating and received financial compensation. Participants were recruited by announcements mails within the University of Tübingen, the University Hospital of Tübingen, and by mouth to mouth propaganda. Individuals with metabolic diseases, including diabetes, cardiovascular or respiratory diseases, or existing pregnancy, were excluded from participation. A medically unremarkable pulmonary function test (spirometry) and electrocardiogram during a bicycle ergometer test until exhaustion were mandatory for study participation. The complete list of in- and exclusion criteria is shown in Appendix A and corresponds to recommended absolute and relative contra-indications for bicycle ergometer testing within occupational medical examinations [24]. The initial bicycle test until exhaustion was also used to determine participants’ individual maximal physical working capacity (PWCmax), with the mechanical power in Watt per kilogram body weight (W/kg) as indicator of cardiorespiratory fitness [25]. Based on PWCmax, participants were categorized in three fitness levels with respect to proposed norm values for PWCmax [26]. Sex-specific categorization was as follows: PWCmax below the norm (men: < 3.0, women: < 2.6 W/kg), corresponding to the norm (men: 3.0 W/kg ≤ x < 4.1 W/kg, women: 2.6 W/kg ≤ x < 3.5 W/kg), and above the norm (men: ≥4.1 W/kg, women: ≥3.6 W/kg).

### 2.2. Study Design, Research Aims and Sample Size

This randomized, intra-subject, cross-over design study included four experimental conditions, each on a separate day, during which the subjects had to accomplish four submaximal bicycle ergometer tests while wearing either no mask (control), a MedMask, and a FFP2exhal or a fabric mask (community mask). The study was registered in the German clinical trial register (DRKS00024531).

The primary aim was to confirm that there are no relevant differences in physical working capacity (PWC) as an indicator of physical performance at a medium level of physical activity (heat rate of 130 beats per minute (bpm)) when wearing a MedMask since wearing a MedMask is the minimal legal requirement for infection prevention in German occupational settings when required distances cannot be guaranteed [2,27].

Secondary, potential performance differences between no mask and MedMask and FFP2exhal and community mask in PWC at the heart rates of 130 and 150 bpm were examined. Complementary, physiological, and subjective responses due to wearing the face masks were analyzed as well as the influence of cardiorespiratory fitness level and sex on potential mask effects.

Sample size was calculated and set according to preliminary results from eight subjects, including four men and four women (Appendix B), assuming equality between MedMask and control at PWC130 during a bicycle ergometer test (primary outcome). Mean differences in PWC130 while wearing MedMask or no mask as well as the corresponding standard deviation were determined. These preliminary data were not included in the final study sample. A relevant effect of the MedMask on PWC130 was considered as a shift in PWC130 to an adjacent fitness level according to published normative values for PWC130 with a change of 0.3 and 0.4 W/kg in females and males, respectively [26]. Using a more conservative approach, 0.3 W/kg was set as relevant change. According to this a sample size, only *n* = 4 would have been necessary for showing noninferiority of the MedMask compared to no mask condition on a statistical significance level of alpha = 0.05. However, we doubted external validity of a study with only 4 subjects. Moreover, due to the secondary research questions and multiple statistical testing, a further increase of sample size was necessary. With a Bonferroni correction of 15, effect sizes of 0.67 could have been shown. Finally, a Williams design preventing first-order carry-over effects with 9 subjects per one of the four randomization sequences (based on four experimental conditions) and balanced for sex and fitness level led to 36 (18 women, 18 men) participants. Due to practical reasons, we had to include three additional participants so that the final sample size was 39 (20 males and 19 females).

### 2.3. Procedures

Participants were invited to an initial visit in the laboratory on which written informed consent was collect, in -and exclusion criteria were verified, and the PWCmax test was conducted. After final inclusion, subjects had to fill out the Nordic Questionnaire and the Physical Activity, Exercise, and Sport Questionnaire. Thereafter, subjects performed the PWCsubmax test on four separate days in randomized order. The first PWCsubmax test took place at least three days after the initial day with the PWCmax test to ensure complete recovery. The PWCsubmax tests were performed at the same time of the day on four consecutive days. Actual intervals between the test conditions can be found in Appendix C. Temperature in the laboratory was kept constant during all measurements at about 23 °C.

### 2.4. Physical Working Capacity Tests and Face Masks

The initial PWCmax test and the submaximal PWC tests (PWCsubmax) were performed on a bicycle ergometer (custo ec5000, custo med GmbH, Ottobrunn, Germany) according to the World Health Organization scheme [28] and followed the updated recommendations for ergometer testing in occupational medicine [24].

PWCmax: At the beginning of the protocol, there was a 3-min phase in which all physiological parameters were measured at rest. Subsequently, cycling started with a resistance of 25 W and cadence of 60 repetitions per minute and increased every 2 min by 25 W until subjective maximal physical exhaustion, or the given cadence could not be maintained. The abort criterion was a decrease of more than 5 repetitions per minute for more than 5 s.

PWCsubmax: The protocol also started with a 3-min rest phase followed by subsequent cycling at 60 repetition per minute with a resistance of 25 W or 50 W. The starting resistance was set according to the PWC150 during the initial PWCmax test. In the case of a PWC150 below 125 W, the starting resistance of 25 W was chosen in order to have more similar cycling durations between subjects. Resistance was also increased by 25 W every two minutes until the level corresponding to at least 70 but no more than 80% of the initial PWCmax was reached. This procedure ensured that every subject reached PWC150 during each submaximal PWC test according to the results of preliminary tests in the 8 subjects also used for sample size calculation. Immediately following both tests (PWCmax, PWCsubmax), the recovery phase began with a 1-min step-out at 25 W and 30 rpm, followed by nine minutes of seated rest. During all ergometer tests, a physician observed subjects’ ECG and health status. Criteria for premature termination of the test can be found in Appendix D. The following face masks were used in this study:MedMask (Figure 1A): Disposable medical face mask without exhaling valve (NITRAS Medical Care Dental GmbH, 4331//PROTECT, medical face mask, made of fiberglass-free non-woven fabric, blue, 3-ply, integrated nosepiece, round and latex-free elastic bands, manufactured according to EN 14683 Type IIRv). This type of mask has been chosen since in Germany it is the minimal legal requirement for infection prevention in occupational settings when the minimum distance cannot be guaranteed [2].FFP2exhal (Figure 1B): Disposable filtering face-piece mask with exhaling valve, protection class II (Honeywell, SuperOne 3206, VALVE, EN 149). This mask was used since suggested wear time limits in Germany for any face mask during the COVID-19 pandemic are based on working conditions where FFP2 masks with exhalation (FFPexhal) valve are used as protective measures against hazardous dust [10].Community mask (Figure 1C): Cotton mouth-nose-cover without exhaling valve (van Laack, Art.49.0946.Z51022.003, CE—2012-16632). A community mask was additionally included as an experimental condition because this type of mask can be used by the general public under the age of 60 and who do not have underlying health conditions according to the recommendation of the WHO [9].

### 2.5. Outcomes and Measurements

Ergometric PWC testing has a long tradition in occupational medicine for assessing whether a sufficiently high level of physical performance for coping with the daily work requirements is given [29]. An imbalance between physical workload and physical work capacity related to aging workers has been suggested to result in chronic overload, increasing the risk of long-term health effects [30,31]. PWC can be tested maximally or submaximally, using performance indicators like VO2max [32] or the mechanical power [33]. In the case of submaximal PWC testing measuring the mechanical power, the achieved power at a given heart rate serves as performance indicator. There are age- and sex-specific norm values [26] that can be used to judge whether differences or changes are within the normal range or can be considered significant. With respect to potential impairments in submaximal performance by wearing a face mask, the assessment of PWC at a certain heart rate level on a bicycle ergometer appears to be an appropriate setting for testing face masks, assuming that increased physiological demands will result in reduced PWC or compensatory mechanisms. In this respect, physiological and subjective responses will serve as important complementary measures.


**Physical working capacity 130 and 150:**


The mechanical power (Watt) during bicycle ergometer testing (custo ec5000, custo med GmbH, Ottobrunn, Germany) was determined with simultaneously measuring the heart rate by a 12-lead electrocardiogram (custo cardio 400, custo med GmbH, Ottobrunn, Germany). Both were recorded at a sample rate of 1 Hz. PWC130 and PWC150 were calculated automatically by the cycle ergometer-ecg system according to linear interpolation. Details will be given in the data analysis section.


**Physiological responses**



*Transcutaneous carbon dioxide partial pressure (tcpCO2)*


A transcutaneous gas measurement device (IntelliVue TcG10, Philips Medical Systems DMC GmbH, Boeblingen, Germany and tc sensor 84, Radiometer GmbH, Krefeld, Germany) was used for non-invasive measurement of carbon dioxide partial pressure at a sample rate of around 0.125 Hz. The assessment of tcpCO2 has already been shown to provide reliable data during exercise testing [34]. The skin sensor electrode with a temperature of 44° Celsius (C) was placed on the right upper arm over the middle deltoid muscle 10 min prior the start of the ergometer test in order to fulfill the warming requirements of this electrode.


*Saturation of peripheral capillary oxygen (SpO2)*


Blood oxygen saturation level represents the amount of oxygen carried in the hemoglobin and is expressed as a percentage of the maximum amount of oxygen that hemoglobin in the blood can carry. Measurements were conducted using an ear pulse oximetry sensor with a sample rate of 0.1 Hz (ES-3227, EnviteC-Wismar GmbH, Wismar, Germany), which was connected to the ergometer system. Using spectrophotometric methodology, pulse oximetry measures oxygen saturation by illuminating the skin and measuring changes in light absorption of oxygenated and deoxygenated blood [35].


*Blood pressure*


Systolic and diastolic blood pressure were measured using a blood pressure cuff applying the riva rochi method at the right arm (blood pressure cuff for ec5000, custo med GmbH, Ottobrunn, Germany). Participants were asked to keep their hand on the handlebars but relax their arm when the measurement took place. The blood pressure cuff was connected to the ergometer system and measured automatically at the end of each load level (sample rate of 0.008 Hz).


*Breathing frequency*


Breathing frequency was measured continuously throughout the test protocols using a sensor belt placed around the chest (NeXus Atem Sensor NEXU2050, Hasomed GmbH, Magdeburg, Germany and AMS42-LAN16fx, BMCM Messsysteme GmbH, Maisach, Germany). The sensor attached to a chest belt recognizes extensions and reductions of the thorax during respiration. The sensor output is a breath-dependent voltage change sampled at the rate of 16 Hz. Breathing frequency was calculated from the end of each load level. However, this interfered with the assessment of perceived exhaustion and perceived respiratory effort, which took place within the last 20 s of each load level. Therefore, and with an additional 5-s buffer, the respiratory rate was calculated from about 30 s before the last 25 s of each load level. Concretely, to calculate the respiratory rate, data from second 63 to second 95 were selected in each 120-s load level. The 512 voltage readings of these 32 s were transferred from the time domain to the frequency domain using Fast Fourier Transform (FFT). The input vector of an FFT must have a length of two to the power of *n*. Thus, in combination with the 16-Hz sampling rate, data from 32 s were used. No windowing and 512 FFT points were used for this purpose. To filter possible interferences (transients), only spectral amplitudes in the frequency range from 0.078 Hz to 1.85 Hz (corresponding to 4.7 breaths/min to 111 breaths/min) were included in the subsequent calculations. A seven FFT point wide moving average filter was used to smooth the frequency-amplitudes for unifying closely spaced spectral local amplitude maxima that may arise due to a gradually increasing respiratory frequency within the 32-s phase. In addition, smoothing reduces occasional peaks caused by motion artifacts. Finally, the mean respiratory rate of each 32-s period corresponds to the frequency of the highest spectral amplitude.


**Subjective responses**



*Respiratory effort and perceived physical exertion*


Respiratory effort and perceived physical exertion were assessed using a modified Borg CR10 scale (0 = nothing at all, 10 = maximal) within the last 20 s of each load level. Although this scale is commonly used for assessing the level of perceived exertion, it is also used for assessing alterations in respiration during bicycle ergometer testing [36].


**Supplemental measurements for characterizing the study sample**


The Nordic questionnaire was used to gather subjects’ age, body weight, gender, laterality, current profession, and weekly working hours [37]. Additionally, the Physical Activity, Exercise, and Sport Questionnaire was used to assess physical activity at work and during leisure time [38].

### 2.6. Data Analysis

Parameter calculation: All outcome variables were assessed at the time points when PWC130 and PWC150 were reached. Due to the incremental test protocol, linear interpolations were necessary to determine the exact point of time and corresponding value for each outcome variable.

Calculation of physical working capacity: The PWC defines the capacity that the subject achieves at the time when it reaches a heart rate of 130 (PWC 130) or 150 (PWC 150) beats per minute. For the calculation of PWC130 and PWC150, the load levels with a minimum duration of 30 s were considered only. Baseline and recovery periods were not included. If the target heart rate was not reached within the last 10 s of a load level, the PWC value was calculated by interpolation or, in case of the last load level, by extrapolation. In the case of extrapolation, the test subject must have reached a heart rate of at least 120 or 140 beats per minute; otherwise, PWC130 or PWC150 were not calculated. The PWC130 and PWC150 values were calculated according to the following Equation:PWC(HR) = W1 + (W2 − W1) × (HR − HR1) / (HR2 − HR1) [W]

R = target heart rate, i.e., 130 or 150

HR1 = mean HR of the last 10 s of the load level before the level in which the target HR was reached in the case of extrapolation: mean HR of the last 10 s of the load level before the level in which the target HR of 120 or 140 was reached, respectively.

HR2 = mean HR of the last 10 s of the load level in which the target HR was reached in the case of extrapolation: HR2 = HR

W1 = watt of the level before the level in which the target HR was reached

W2 = watt of the level in which the target HR was reached

In addition, the PWC related to the body weight was calculated:$$PWC_{rel}\left(HR\right)=\frac{PWC\left(HR\right)}{body\;weight}\;\left[W/kg\right]$$

Calculation of outcomes at the time points when PWC130 and PWC150 were reached: For the variables peripheral capillary oxygen saturation (SpO2), systolic and diastolic blood pressure (SBP and DBP), and transcutaneous carbon dioxide partial pressure (tcpCO2), a linear regression was performed around the time when PWC130 or PWC150 were reached. The regression included as many as possible but at least two of the measured values belonging to the respective variable. Only values recorded during the exercise period were considered; baseline and recovery periods were not included in the regression. SpO2 was considered over an interval of 40 s around the time of PWC with a maximum of 5 values (sampling interval of approximately 10 s). SBP and DBP were determined over an interval of 260 s around the time of PWC with a maximum of 3 values (sampling interval of approximately 120 s) considered. Linear regression of tcpCO2 was performed over an interval of approximately 45 s around the time of PWC with a maximum of 6 values (sampling interval of approximately 7–9 s).

For the variables breathing frequency, respiratory effort and perceived physical exertion a linear increase was assumed at each 2-min load level when the respective power of the cycling protocol was reached (end of a load level). For the calculation of the linear regression, all measured values of the parameter from the beginning of the measurement to the end of the exercise period (sampling interval of approximately 120 s) were considered. Values were than taken from the corresponding time points of PWC130 and PWC150.

Statistical analysis: The primary hypothesis was to show noninferiority of the MedMask condition in comparison to not wearing a mask. This was done by calculating the two-sided 95% confidence interval of the difference in the primary endpoint (PWC130) between control condition and MedMask (regression coefficient for type of mask in a linear mixed model including only these two conditions, coded as 0 and 1, and sex).

For all secondary aims of this study, linear mixed models (LMM) were applied to analyze the influence of the independent variables mask condition, sex, cardiorespiratory fitness (PWCmax), the interaction of mask condition with sex, and mask condition with cardiorespiratory fitness on the dependent variables (primary outcome: PWC130; secondary outcomes: PWC150, additional parameters at the time points when PWC130 and PWC150 were reached, i.e., tcpCO2, SpO2, systolic and diastolic blood pressure, breathing frequency, perceived respiratory effort, and perceived physical exertion). Therefore, five variations of the LMM were performed per outcome variable. The first LMM included only mask condition as independent variable. The second LMM additionally included sex, and the third LMM included sex and cardiorespiratory fitness. LMM four and LMM five additionally included the interaction terms mask condition x sex and mask condition x cardiorespiratory fitness. Alpha level was Bonferroni corrected for 15 comparisons. All outcome variables were visually inspected for extreme values and normal distribution. Means and standard deviations or boxplots, including median and the upper and lower quartiles or frequencies, were used to describe the results. Differences in anthropometric data between the three fitness level groups were analyzed using a one factorial Analysis of Variance (ANOVA) and Tukey’s honestly significant differences test for post-hoc comparison. The statistical software IBM^®^ SPSS^®^ Statistics Version 27 was used for statistical analyses.

## 3. Results

### 3.1. Dropouts

Nineteen subjects were not included after the initial visit. In 18 cases, the reason for not being included was that the presumed fitness level (based on participants self-report) was not reached during the PWCmax test. These subjects could not be included in the study because the required number of participants at the next lower or higher fitness level had already been fully recruited. One subject refused to participate after the initial visit due time constraints. No participant dropped out for medical reasons or due to adverse events during the measurements.

### 3.2. Characteristics of the Final Study Sample

Participants were 38.2 years old, had a BMI of 23.7 kg/m², and their length of employment in the current occupational profession was 10.5 years. Furthermore, participants had a weekly working time of 34 h and reported their weekly physical activity at work and at leisure time being 492 min and 206 min, respectively. Only four participants were smokers. A detailed overview of the characteristics of the final study sample is provided in Table 1. Participants with a high cardiorespiratory fitness level (level 3) were younger in age, had a shorter duration in their current profession, and were more active during leisure time then the participants from the other two fitness levels. Furthermore, BMI was lower than from participants from fitness level 1.

### 3.3. Normal Distribution and Missing Data

After visual inspection of the outcome variables an acceptable level of normal distribution based on histograms, skewness and kurtosis (between –1 and 1) could be assumed. Only for the variable SpO2 did an obvious violation of normal distribution occur. The amount of missing data varied between variables and time point of assessment. Generally, there were fewer missing data at the time point of PWC130 than at PWC150, and the primary outcome variable physical performance at the heart rate of 130 bpm had the lowest amount of missing data (≤5%). The highest amount of missing data occurred in the parameters tcpCO2, SpO2, and blood pressure. A comprehensive overview of missing data per variable is given in Appendix E. The applied statistical analysis (linear mixed models) are considered being robust against violation of normal distribution [39] and able to deal with missing data without causing problems with power and bias when missing data are at random [40].

### 3.4. Primary Outcome—Physical Working Capacity at the Heart Rate of 130 Beats per Minute Using a MedMask

The confidence limit for the difference between the MedMask and the no mask condition, adjusted for sex, was −0.058, 95% CI −0.188 to 0.002. Thus, the noninferiority criterion was met, and the study could successful prove noninferiority of the MedMask vs. the control situation without mask (Table 2). Statistical models without sex and models including sex and cardiorespiratory fitness did not change the differences between mask conditions or standard errors or CIs. Furthermore, no interaction of mask condition with sex and mask condition with cardiorespiratory fitness occurred. All statistical models can be found in Appendix F.

### 3.5. Secondary Outcomes

**Physical working capacity at a heart rate of 130 and 150 bpm using a MedMask, FFP2exhal, and community mask:** Using the MedMask did not lead to a statistically significant change in PWC150 vs. control. Again, sex and cardiorespiratory fitness level had no effects, as seen from LMM2 to LMM5 (Appendix F). Wearing the community mask was also not associated with any statistically significant changes in PWC130 or PWC150 vs. control. When FFP2exhal was used, a slightly lower PWC130 value was found compared to control (FFP2exhal: −0.17 W/kg, *p* < 0.0001) without exceeding the a-priori defined relevant change of 0.3 W/kg. No change in PWC occurred at the heart rate of 150 bpm. Adding sex and cardiorespiratory fitness to the model (LMM 2 and LMM3) did not induce further changes, and no interaction of mask condition with sex and mask condition with cardiorespiratory fitness occurred (LMM4 and LMM5). Boxplots of PWC130 and PWC150 resulting from the four experimental conditions are shown in Figure 2. Means and standard deviations from all outcomes and experimental conditions are shown in Table 3.

**Physiological responses at PWC130 and PWC150:** None of the investigated face masks led to statistically significant changes in the physiological response at PWC130 or PWC150 compared to the control condition without mask. Neither sex nor cardiorespiratory fitness level influenced physiological responses due to mask application. Figure 3, Figure 4, Figure 5 and Figure 6 include tcpCO2, SpO2, systolic and diastolic blood pressure, and breathing frequency in response to the four experimental conditions.

**Subjective responses at PWC130 and PWC150:** The perceived respiratory effort was higher when masks were worn (Figure 7). The mean difference between the control condition and the MedMask was 0.6 points (CI 0.3 to 0.9, *p* < 0.0001). In comparison to not using a face mask, the community mask and FFP2exhal led to a mean increase of 0.8 (CI 0.5 to 1.1, *p* < 0.0001) and 1 point (CI 0.7 to 1.3, *p* < 0.0001) on the zero-to-ten Borg-CR10 scale, respectively. The perceived level of physical exertion did not change when wearing any of the three face masks (Figure 8).

## 4. Discussion

The primary aim of the present study was to confirm that wearing a MedMask does not decrease physical working capacity (PWC). Secondary, the influence of a filtering facepiece mask with exhalation valve class 2 (FFP2exhal) and a cotton fabric mask (community mask) on PWC was also investigated. Complementary, corresponding physiological and subjective responses as well as a potential moderating role of subjects’ cardiorespiratory fitness and sex on mask effects were analyzed.

Noninferiority of the MedMask in PWC130 compared to not wearing a mask could be proved. Additionally, the results clearly indicated that wearing typical face masks recommended during the COVID-19 pandemic does not influence PWC, as was shown by the mechanical power in watt per kilogram bodyweight at the heart rates 130 and 150 bpm during a standardized bicycle ergometer protocol in a relevant manner although PWC130 was slightly and statistically significant reduced when using the FFP2exhal compared to the no mask condition. Neither was wearing a MedMask nor a FFP2exhal or community mask associated with any relevant changes in physiological response. However, respiratory effort has been perceived somewhat more demanding when wearing face masks. Sex and cardiorespiratory fitness level did not change the influence of the face masks.

In many occupational settings, wearing a MedMask has become mandatory, and depending on the circumstances at work, different face masks are required. Many types of masks have been shown to have a high potential in reducing the risk for aerosol transmission of SARS-CoV-2 when properly used [4]. Potential negative side effects of the masks on wearers’ physical performance and health were already addressed by several studies summarized by Kisilinsky and colleagues [8]. However, the currently available literature is considered to lacking high-quality data from well-design studies [15]. The present well-powered study with its randomized intra-individual cross-over design and 39 participants (grouped in three cardiorespiratory fitness levels, mixed age structure, balanced for sex) aims to address this shortcoming in the scientific literature.

Several studies already tested potential side effects of face masks using different bicycle ergometer protocols and outcomes [13,14,16]. Fikenzer, Uhe, Lavall, Rudolph, Falz, Busse, Hepp, and Laufs [14] applied a ramp protocol until maximum exertion with an increase of 50 W within 3 min. In the study by Georgi and colleagues, three load levels (50, 75, and 100 W) were applied for a period of 3 min, and steady state cycling for 30 min was used [13]. Furthermore, Fikenzer et al. and Lässing et al. measured pulmonary function parameters by putting an additional spirometry mask over the face mask. Both found increased physiological demands when wearing a MedMask, and Fiekenzer et al. reported additional impairments by using a FFP2/N95 mask [13,14]. This approach has significant limitations in terms of generalizability since the typical leakage characteristics under daily use has been changed. Georgi, Haase-Fielitz, Meretz, Gasert, and Butter [16] did not use measurements interfering with the face masks and found clinically non-relevant changes in tcpCO2 when wearing either a MedMask, FFP2, or fabric mask compared to the no mask. SpO2 was slightly reduced when a FFP2 mask was worn but again in a clinically non-relevant range. Our study revealed no changes in tcpCO2 or SpO2 between mask and control conditions. Of course, our experimental approach was different from Georgi et al. since we did not use absolute load levels. However, our results also showed no change in the individual mechanical power when a MedMask or community mask were worn. Minor reductions in PWC130 occurred when the FFP2exhal was used, but these changes of 0.17 W/kg can also be regarded as non-relevant related to proposed norm values [26]. Furthermore, we used a FFP2 mask with exhalation valve instead of a common FFP2/N95 mask. By using a FFP2exhal, carbon dioxide rebreathing can be avoided by a better breathing gas exchange during the exhalation phase, which may be the main reason for discrepancies between our results and the results of Georgi et al. [16]. The reason for using an FFP2exhal in the present study are the German recommendations for wear-time limits at the workplace, which will be discussed in a later paragraph of the paper. Generally, our study confirms many results of previously conducted studies. We did not find any relevant change in physical performance and physiological response when wearing a face mask. This was also shown by Ramos-Camp, et al. [41], who reported similar strength performance and physiological response when using a surgical or FFP2 mask during resistance training in people with sarcopenia. In addition, the authors of a recent review article [15] concluded that although dyspnea may be increased, and perceived effort may be altered when using face masks, the effects on work of breathing, blood gases, and other physiological parameters during physical activity are small, often too small to be detected. This conclusion exactly represents the results of the present study with higher self-reported respiratory effort when wearing a mask but no effects on breathing frequency or any other physiological response. Additionally, no conclusions for moderating factors, such as sex-based differences in the physiological responses to exercise while wearing a face mask, could be drawn in the review [15]. To investigate the potential moderating role of sex and cardiorespiratory fitness was a secondary study outcome. In this respect, neither sex nor cardiorespiratory fitness were identified as moderating factors of mask effects. A previous study indicated an association of a low cardiorespiratory fitness level with more negative responses when wearing face masks [16], which could not be confirmed by the present study.

As already mentioned in the introduction section, scientific results regarding potential side effects of face masks are urgently required not only for improving scientific knowledge but for providing a scientific basis for face mask recommendations during the COVID-19 pandemic. In Germany, wear-time limitations for masks worn for COVID-19 prevention at the workplace have been adopted from wear-time limits to avoid overload of the users when using particle-filtering half-masks with an exhalation valve (similar to N95 masks with respect to the respiratory resistance and the total leakage) as protective measure against hazardous dust [10]. It is further recommended to adjust wear-time limits based the required physical workload at a workplace. In this context, a low physical workload would allow up to three hours of mask use, and high workloads would further reduce the recommended wearing time of two hours [11]. Although our study was not directly designed to investigate the effects of different wearing times, the results of the present study do not provide any reason to support this recommendation. There were no additional physical demands when wearing face masks at a medium physical activity (heat rate of 130 bpm) resulting in lower PWC values. The continuous exercising without removing the masks until the heart rate of 150 bpm did also not lead to any impairments regarding PWC or negative physiological responses. Another recommendation regarding wearing-time limitations of face masks at the workplace, which is in better agreement with the results of the present study, is given by the German Federal Institute of Occupational Safety and Health. For medical masks, no health hazards are assumed [42]; for FFP2 masks, individual and occupational activity-dependent regulations should be made according to a risk assessment at the workplace and in consultation with an occupational physician [43].

### Limitations

A limitation of the present study is the rather short observation period. In a study with a 30-min steady-state bicycle ergometer protocol, changes in heart rate due to wearing a MedMask only appeared at the end of the 30-min period [13]. Although Lassing, Falz, Pokel, Fikenzer, Laufs, Schulze, Holldobler, Rudrich, and Busse [13] applied a spirometry mask over a surgical mask, leading to an unrealistic leakage, it could be that changes may occur after prolonged face mask use. Another limitation to be mentioned is an increasing number of missing data in SpO2, tcpCO2, and blood pressure (Appendix E) with increasing workload. Although all methods are used during exercise [34,44,45], data loss was unsatisfying. However, the relatively large sample size and the fact that data losses occurred in all experimental conditions, as well as a statistical analysis able to deal with missing data, allow us to consider the results reliable. Finally, psychological aspects of face-mask wearing were not considered.

## 5. Conclusions

The results of the present study provide reason to believe that wearing face masks for infection prevention during the COVID-19 pandemic does not pose relevant additional physical strain on the user although some more respiratory effort is required.

## Figures and Tables

**Figure 1 ijerph-19-01063-f001:**
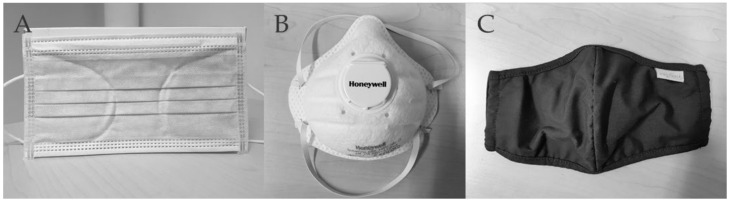
Investigated face masks. (**A**) MedMask medical face mask (NITRAS PROTECT, EN14683), (**B**) FFP2exhal filtering facepiece with exhalation valve (Honeywell, SuperOne 3206, VALVE, EN 149) and (**C**) community mask fabric mask (van Laack, CE—2012–16632).

**Figure 2 ijerph-19-01063-f002:**
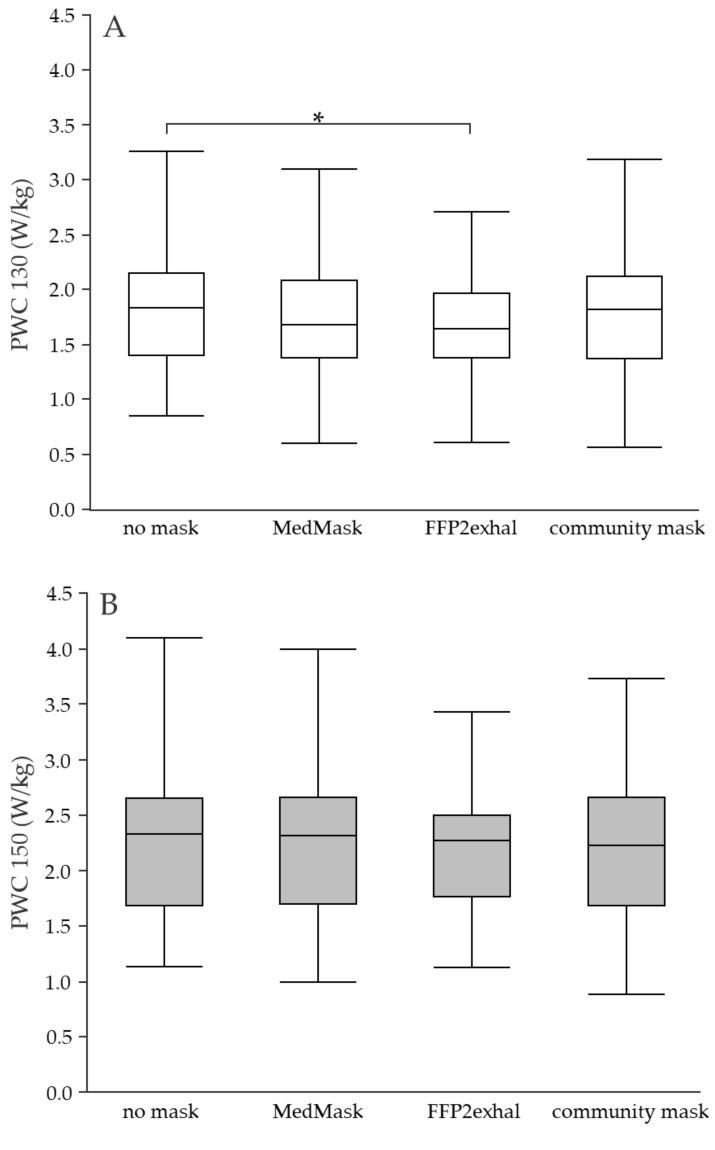
(**A**,**B**) Physical working capacity at the heart rate of 130 and 150 beats per minute using different face masks. Grey shaded boxplots represent the physical working capacity at the heart rate of 150 beats per minute. Asterisks indicate statistically significant differences (Bonferroni corrected *p* < 0.003). PWC, physical working capacity; MedMask, medical face mask; FFP2exhal, filtering facepiece with exhalation valve.

**Figure 3 ijerph-19-01063-f003:**
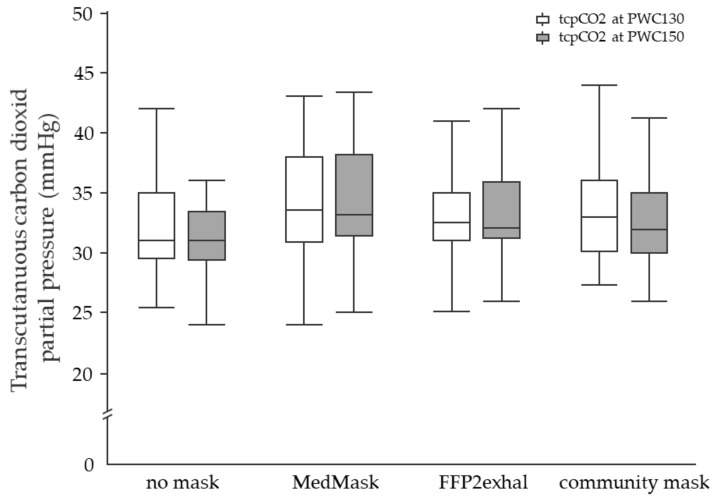
Transcutaneous carbon dioxide partial pressure at the heart rate of 130 and 150 beats per minute using different face masks. Grey shaded boxplots represent the tcpCO2 at PWC150. tcpCO2, transcutaneous carbon dioxide partial pressure; PWC, physical working capacity; MedMask, medical face mask; FFP2exhal, filtering facepiece with exhalation valve.

**Figure 4 ijerph-19-01063-f004:**
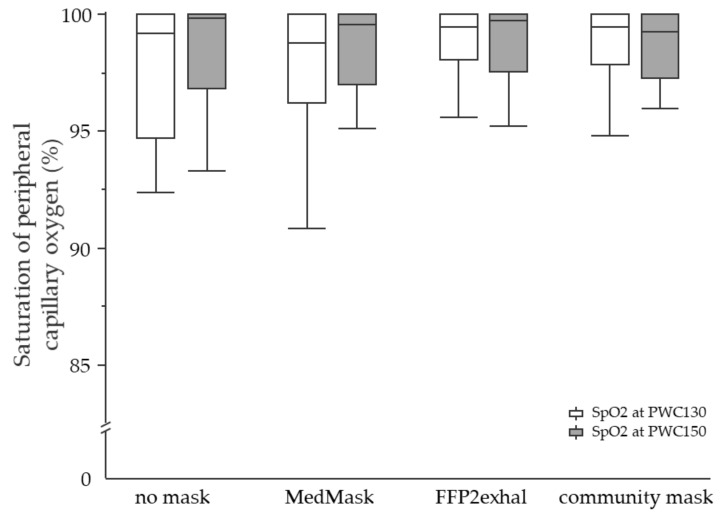
Saturation of peripheral capillary oxygen at the heart rate of 130 and 150 beats per minute using different face masks. Grey shaded boxplots represent oxygen saturation at PWC150. SpO2, saturation of peripheral capillary oxygen; PWC, physical working capacity; MedMask, medical face mask; FFP2exhal, filtering facepiece with exhalation valve.

**Figure 5 ijerph-19-01063-f005:**
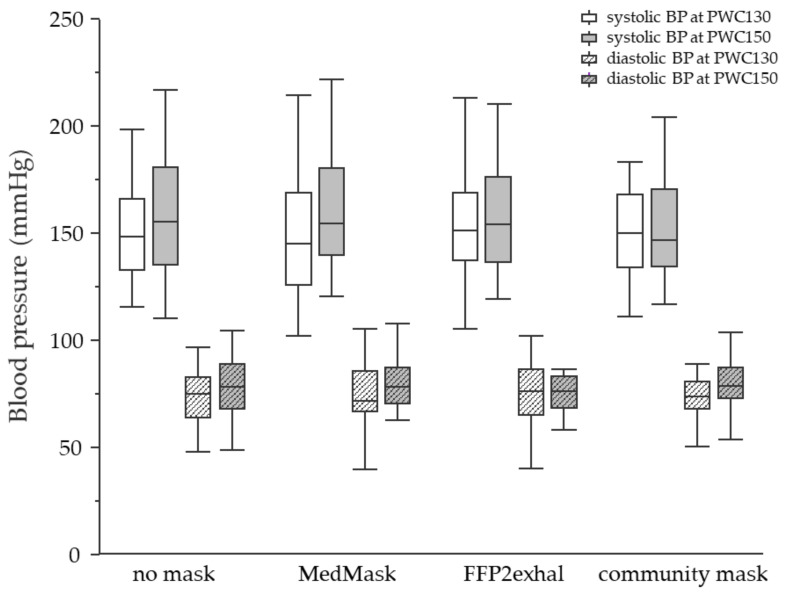
Systolic and diastolic blood pressure at the heart rate of 130 and 150 beats per minute using different face masks. Filled boxplots represent the systolic blood pressure and grey shaded boxplots represent blood pressure at PWC150. SpO2, saturation of peripheral capillary oxygen; PWC, physical working capacity; MedMask, medical face mask; FFP2exhal, filtering facepiece with exhalation valve.

**Figure 6 ijerph-19-01063-f006:**
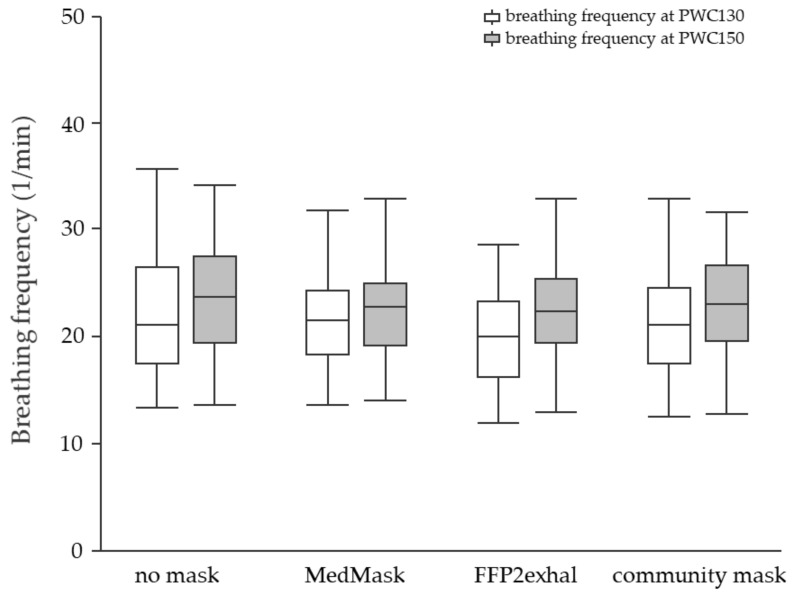
Breathing frequency at the heart rate of 130 and 150 beats per minute using different face masks. Grey shaded boxplots represent breathing frequency at PWC150. PWC, physical working capacity; MedMask, medical face mask; FFP2exhal, filtering facepiece with exhalation valve.

**Figure 7 ijerph-19-01063-f007:**
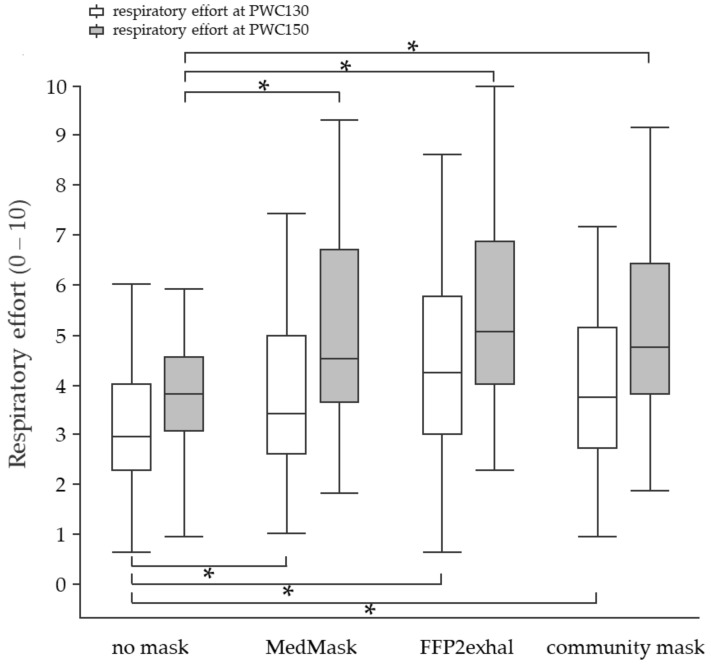
Perceived respiratory effort at the heart rate of 130 and 150 beats per minute using different face masks. Grey shaded boxplots represent respiratory effort at PWC150. Asterisks indicate statistically significant differences (Bonferroni corrected *p* < 0.003). 0 = no respiratory effort, 10 = maximum respiratory effort. PWC, physical working capacity; MedMask, medical face mask; FFP2exhal, filtering facepiece with exhalation valve.

**Figure 8 ijerph-19-01063-f008:**
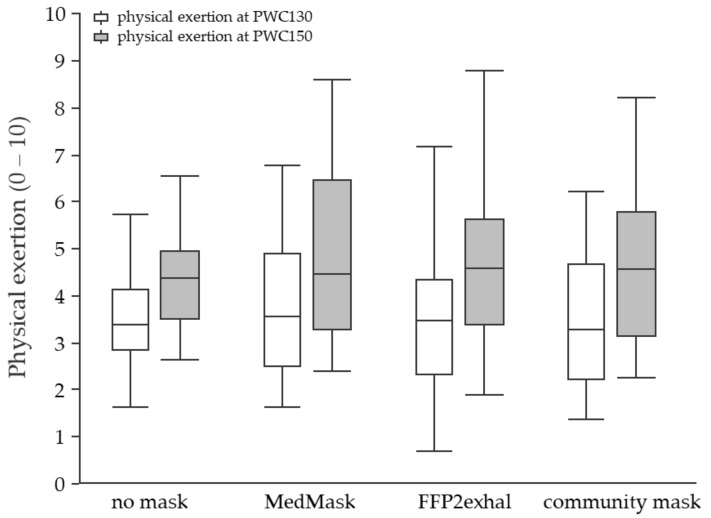
Perceived physical exertion at the heart rate of 130 and 150 beats per minute using different face masks. Grey shaded boxplots represent physical exertion at PWC150. 0 = no physical exertion, 10 = maximum physical exertion. PWC, physical working capacity; MedMask, medical face mask; FFP2exhal, filtering facepiece with exhalation valve.

**Table 1 ijerph-19-01063-t001:** Characteristics of the study sample.

Parameter			Overall	Fitness Level 1	Fitness Level 2	Fitness Level 3
PWCmax	(W/kg)	mean SD	3.3 1.0	2.3 * 0.3	3.2 * 0.3	4.4 * 0.6
sex	(n, %)	overall men women	39, 100% 20, 51.3% 19, 48.7%	13, 33.3% 6, 15.4% 7, 18.0%	13, 33.3% 7, 18.0% 6, 15.4%	13, 33.3% 7, 18.0% 6, 15.4%
smokers	(n, %)		4, 10.3%	1, 2.6%	3, 7.7%	0, 0%
age	(years)	mean SD	38.2 14.2	43.1 16.9	43.8 13.1	28.7 ** 6.1
BMI	(kg/m²)	mean SD	23.7 2.4	25.0 3.1	23.5 1.9	22.5 * 1.4
Duration of current occupational profession	(years)	mean SD	10.5 10.7	13.8 13.5	12.9 9.9	5.0 ** 5.5
Weekly working time	(hours)	mean SD	34.3 13.2	30.5 14.2	32.6 11.3	39.7 13.2
Physical activity at work	(minutes/week)	mean SD	491.8 589.0	424.3 488.1	609.3 726.6	441.7 556.5
Physical activity leisure time	(minutes/week)	mean SD	206.4 207.3	99.1 103.7	112.3 100.2	407.7 ** 222.2

PWCmax, physical working capacity (mechanical power, Watt/kg) during a bicycle ergometer test until maximum exhaustion; BMI, body mass index; ** statistically significant difference to the other two fitness level groups (*p* < 0.05), * statistically significant difference to fitness level group 1 (*p* < 0.05).

**Table 2 ijerph-19-01063-t002:** Comparison between the no mask and MedMask condition.

Model	Outcome Parameter	Dependent Variable	Degree of Freedom	F-Value	*p*-Value
1	PWC130	Face mask condition	1	3.472	0.07
2	PWC130	Face mask condition	1	3.491	0.07
		sex	1	13.245	0.001

**Table 3 ijerph-19-01063-t003:** Mean values of all assessed outcome parameters during the four experimental conditions.

Outcome Parameter	No Mask		MedMask		FFP2exhal		Community Mask	
	Mean	SD	Mean	SD	Mean	SD	Mean	SD
PWC130 (W/kg)	1.91	0.70	1.84	0.72	**1.74**	**0.65**	1.83	0.70
PWC150 (W/kg)	2.35	0.83	2.32	0.81	2.28	0.76	2.35	0.89
tcpCO2 at PWC130 (mmHg)	32.37	6.22	34.07	4.28	33.64	4.29	33.70	4.48
tcpCO2 at PWC150 (mmHg)	31.18	6.15	34.33	4.70	33.45	4.54	32.78	4.29
SpO2 at PWC 130 (%)	97.30	3.74	97.42	3.38	98.29	2.94	97.90	4.03
SpO2 at PWC 150 (%)	97.45	4.84	98.11	3.02	98.66	2.17	98.18	2.71
Systolic blood pressure at PWC130 (mmHg)	151.00	28.67	149.34	29.18	152.38	29.30	150.79	25.75
Systolic blood pressure at PWC150 (mmHg)	157.45	28.28	160.80	28.54	155.11	24.18	152.51	24.34
Dyastolic blood pressure at PWC130 (mmHg)	75.19	13.81	74.76	14.37	76.13	15.84	74.32	10.57
Dyastolic blood pressure PWC150 (mmHg)	78.61	14.48	77.88	14.15	77.02	12.24	79.22	11.68
Breathing frequency at PWC130 (1/min)	22.34	5.79	21.60	4.59	21.76	10.97	21.38	4.53
Breathing frequency at PWC150 (1/min)	24.15	6.64	23.09	4.74	22.26	4.87	23.00	4.67
Perceived respiratory effort at PWC130 (0–10)	3.19	1.52	**3.89**	**1.70**	**4.30**	**1.92**	**3.96**	**1.59**
Perceived respiratory effort at PWC150 (0–10)	3.97	1.71	**5.12**	**1.92**	**5.34**	**1.95**	**5.12**	**1.83**
Perceived physical exertion at PWC130 (0–10)	3.51	1.25	3.80	1.67	3.59	1.83	3.42	1.39
Perceived physical exertion at PWC150 (0–10)	4.47	1.35	4.87	1.70	4.59	1.83	4.51	1.66

Bold and grey shaded numbers indicate statistically significant differences in comparison to the no mask condition (Bonferroni corrected *p* < 0.003). tcpCO2, transcutaneous carbon dioxide partial pressure; SpO2, saturation of peripheral capillary oxygen; PWC, physical working capacity; MedMask, medical face mask; FFP2exhal, filtering facepiece with exhalation valve.

## Data Availability

The data are not publicly available due to data use restrictions contained in study participants’ information material.

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
