# Peer review of "Effects of Face Masks on Physical Performance and Physiological Response during a Submaximal Bicycle Ergometer Test"

_ijerph, 2022, doi:10.3390/ijerph19031063_

Round 1

Reviewer 1 Report

Thanks for the excellent work! Just curious about the recruitment of the subjects: what was the response rate since the team uses word of mouth and announcement mails, to hit the minimum target of 39? No mention about the order of the masks for each subject, was the order random or every subject got the same sequence of masks?

Author Response

Dear reviewer,

thank you very much for your positive feedback. The response rate was very high. When we announced this study via email more than enough interested immediately people replied. However, it was difficult to recruit the group with the high cardiorespiratory fitness level, because many of the subjects overestimated themselves. As we mentioned in the dropout section we had 19 additional subjects who took part in a PWCmax test who then could not be included before we reached our final sample. Furthermore, we used a Williams design with four randomization order sequences (according to the four mask conditions) with 9 subjects per sequence, so our target sample size actually was 36 but for practical reasons we had three subjects in addition. We also described this in the method section.

Reviewer 2 Report

Abstract

Toward the end – the conclusion: “Using the MedMask…  … did not influence mask effects”. This section could be simplified. The results were good, but this just overcomplicates the findings.

Intro

The penultimate paragraph mentions that you are looking for differences in PWC. Differences between what? Be clear here so the reader knows. I think you should simplify and state that the first aim was to see if wearing a mask impairs performance by looking at different variables during a submaximal exercise intensity. Then, I suggest removing the final paragraph as it’s getting into the details of the methodology.

Methods

Section 2.3 – was the test stopped if the cadence dropped below a certain value?

Why was the PWC130/150 chosen? It seems as though you could have just chosen certain percentages of their maximal work capacity, rather than work capacity at certain submaximal HRs. I feel like more explanation on why you chose those specifically is warranted.

Pictures of the masks would be beneficial for the reader.

Suggest including the demographics in this section – and include the mean, SD and range for the variables.

Why is 32s used for the breathing frequency interval?

Why was there such a large range between testing sessions? It looks like some subjects tested on subsequent days and others had nearly two weeks between. Was their fitness/exercise monitored during that time period? How do you know if their fitness stayed at the same level?

Why was a randomized test order chosen? With four conditions, a balanced Latin square would have been very useful.

Results

This section (along with appendices section) need a major revision. I highly suggest a more concise presentation of the results (do not need all of those appendices). Way too many appendices, tables and figures. Simplify!

Otherwise it is a good study with very interesting (and useful) results. Well done.

Author Response

Dear reviewer,

thank you very much for your comments and suggestions which were very useful to improve our manuscript.

Abstract

Toward the end – the conclusion: “Using the MedMask…  … did not influence mask effects”. This section could be simplified. The results were good, but this just overcomplicates the findings.

  • thank you for your comment. We have revised the text in order to make things clearer.

Intro

The penultimate paragraph mentions that you are looking for differences in PWC. Differences between what? Be clear here so the reader knows. I think you should simplify and state that the first aim was to see if wearing a mask impairs performance by looking at different variables during a submaximal exercise intensity. Then, I suggest removing the final paragraph as it’s getting into the details of the methodology.

  • In fact, we have formulated the study aims in a somewhat complicated way. We reformulated the study aim in a more general way and moved the two specific study aims to the method section since our sample size calculation refers to the primary study outcome (performance between MedMask and control).
    New text in the introduction: “Therefore, this study investigated whether face masks would impair physical performance and affect physiological and subjective response during submaximal physical activity as well as a potential moderating role of cardiorespiratory fitness level and sex on the mask effects.”
    Text moved to the method section 2.2 Study design, research aims and sample size: “The primary aim was to confirm that there are no relevant differences in physical working capacity (PWC), as an indicator of physical performance, at a medium level of physical activity (heat rate of 130 beats per minute (bpm)) when wearing a MedMask since wearing a MedMask is the minimal legal requirement for infection prevention in German occupational settings when required distances can´t be guaranteed [2,24].
    Secondary, potential performance differences between no mask and MedMask, FFP2exhal and community mask in PWC at the heart rates of 130 and 150 bpm were examined. Complementary, physiological and subjective responses due to wearing the face masks were analyzed as well as the influence of cardiorespiratory fitness level and sex on potential mask effects.”

Methods

Section 2.3 – was the test stopped if the cadence dropped below a certain value?

  • When the cadence dropped more than 5 rpm for more than 5 seconds the test was stopped. We slightly adjusted the original text in the paragraph PWCmax (now in section 2.4) and added the abort criterion: “…or the given cadence could not be maintained. The abort criterion was a decrease of more than 5 repetitions per minute for more than 5 seconds.”

Why was the PWC130/150 chosen? It seems as though you could have just chosen certain percentages of their maximal work capacity, rather than work capacity at certain submaximal HRs. I feel like more explanation on why you chose those specifically is warranted.

  • We chose the concept of PWC130 and PWC150 because of the available norm values which allowed us to define a relevant change in performance and therefore to calculate the required sample size. We gave more information about why choosing PWC in section 2.5 (new text in bold letters): Ergometric PWC testing has a long tradition in occupational medicine for assessing whether a sufficiently high level of physical performance for coping with the daily work requirements is given (Sammito et al. 2020). An imbalance between physical workload and physical work capacity related to aging workers has been suggested to result in chronic overload, increasing the risk of long-term health effects [29,30]. PWC can be tested maximal or submaximal using performance indicators like VO2max [32] or the mechanical power [33]. In the case of submaximal PWC testing measuring the mechanical power, the achieved power at a given heart rate serves as performance indicator. There are age and sex specific norm values [27] which can be used to judge whether differences or changes are within the normal range or can be considered significant. With respect to potential impairments in submaximal performance by wearing a face mask the assessment of PWC at a certain heart rate level on a bicycle ergometer appears to be an appropriate setting for testing face masks assuming that increased physiological demands will result in reduced PWC or compensatory mechanisms. In this respect, physiological and subjective responses will serve as important complementary measures.

Pictures of the masks would be beneficial for the reader.

  • Pictures of the masks were included in Figure 1.

Suggest including the demographics in this section – and include the mean, SD and range for the variables.

  • Demographics are given at the beginning of the results section.

Why is 32s used for the breathing frequency interval?

  • Our aim was to calculate the breathing frequency from the end of each load level. According to our preliminary testings a duration of about 30 seconds would be sufficient to provide a solid data basis (not influence by single extreme values) to calculate berating frequency. However, due to the requirements of a fast furrier transformation 32s had to be used. We provided this information in the text as follows (new text in bold letters): Breathing frequency was calculated from the end of each load level. However, this interfered with the assessment of perceived exhaustion and perceived respiratory effort, which took place within the last 20 s of each load level. Therefore, and with an additional 5 s buffer, the respiratory rate was calculated from about 30 s before the last 25 s of each load level. Concretely, to calculate the respiratory rate, data from second 63 to second 95 were selected in each 120-second load level. The 512 voltage readings of these 32 seconds were transferred from the time domain to the frequency domain using Fast Fourier Transform (FFT). The input vector of an FFT must have a length of two to the power of n. Thus, in combination with the 16 Hz sampling rate data from 32 s were used.

Why was there such a large range between testing sessions? It looks like some subjects tested on subsequent days and others had nearly two weeks between. Was their fitness/exercise monitored during that time period? How do you know if their fitness stayed at the same level?

  • Actually, the range between testing sessions was rather small with a median of 1 day between the four submaximal testing sessions and a median of 4 days between the max test and the first submaximal session. Only for a few subjects longer intervals occurred. We added the 25th and 75th percentile of time intervals in table E1 to provide a clearer picture. Of course, we cannot be sure that the fitness level changed in those who had a long break between PWCmax and the submaximal tests. However, the time intervals between the submaximal tests were shorter so that potential differences between the results of the submaximal tests should really depend on the masks and not on changes in fitness level.

Why was a randomized test order chosen? With four conditions, a balanced Latin square would have been very useful.

  • A full factorial Latin square design with four conditions and three fitness groups would have led to 72 subjects. As our sample size calculation resulted in much smaller sample size, we used a Walliams design which controls for first order carry over effect balanced for sex and fitness level. We already mentioned this in the method section.

Results

This section (along with appendices section) need a major revision. I highly suggest a more concise presentation of the results (do not need all of those appendices). Way too many appendices, tables and figures. Simplify!

  • Dear reviewer, we have read the results again thoroughly and looked for ways to simplify them. In our opinion, the given division into primary and secondary outcomes already structures the results section very well and we would like to refrain from making further adjustments here, especially since none of the other two reviewers commented on this. Based on the comment of another reviewer we highlighted all significant differences in Table 3 and in the corresponding figures which give a clearer picture of the major results.

Otherwise it is a good study with very interesting (and useful) results. Well done.

  • Thank you very much we appreciate this assessment.

Reviewer 3 Report

It is useful for a reviewer for you to include line numbers. These seem to suddenly be introduced half way through the paper.

Abstract. A brief sentence of rationale for the study would be useful.

Abstract. Include units when referring to heart rate.

Abstract. What does relevant changes mean? Does it mean there was no statistically significant difference? If so, then just state this. If not, then state what is does mean.

Introduction. The reviews you cite, references 7 and 8, are not really comparable. One was specifically looking at exercise and the other more generally. In the case of this study, the Shaw review is the most relevant for recreational exercise. The Kisilinsky review might be more relevant for occupational exercise. It is not clear whether recreational or occupational exercise is the main focus of this paper.

Method. What is the rationale for using PWC130 and 150. Why not measure a fixed power output or % power output are record heart rate? Given the criticism of the Shaw review for focusing on maximal exercise, your choice of a specific submaximal exercise needs expanding on. Appendix D should be put in the method section to provide more detail on PWC130 and 150.

Method. First mention of a warm-up prior to exercise comes at the end of the first physiological responses paragraph. Warm-up should be described in detail earlier on in methods, when talking about the maximal and submaximal tests performed.

Method. Why was respiratory rate calculated from 63-95 seconds of each stage? Those sound like oddly specific numbers.

Method. Were respiratory effort and perceived exertion measured separately, but using the same scale? If so, why? Is there not a more appropriate/specific scale that could be used for respiratory effort?

Method. It seems strange that the procedures section comes so far into the method. I would have thought this should be one of the first things discussed.

Results. I don't find the dropout section relevant.

Results. Some statistical analysis to support the statements regarding differences between groups for age, BMI and physical activity would be useful.

Table 3. Please highlight any statistically significant differences within the table. The same goes for the figures.

Figure 2. The legend for figure 2 is in German. Should at tcpCO2 at PWC130, rather than bei PWC130.

All captions for tables and figures seem to have the manufacturer of the mask listed as well. This is not necessary as this was given at the first mention of each of the mask.

Author Response

Dear reviewer,

thank you very much for your comments and suggestions which were very useful to improve our manuscript.

It is useful for a reviewer for you to include line numbers. These seem to suddenly be introduced half way through the paper.

  • We are very sorry to hear about problems with line numbering. However, line numbers were created automatically by the submission system (or not). We will inform the editorial office about this problem.

Abstract. A brief sentence of rationale for the study would be useful.

  • Thank you for this suggestion. We added the following sentence to the beginning of the abstract (new text in bold letters): The ongoing COVID-19 pandemic requires wearing face masks in many areas of our daily life and potential side effects of mask use is discussed. Therefore, the present study explores whether wearing a medical face mask (MedMask) affects physical working capacity (PWC)”.

Abstract. Include units when referring to heart rate.

  • We added beats per minute

Abstract. What does relevant changes mean? Does it mean there was no statistically significant difference? If so, then just state this. If not, then state what is does mean.

  • We changed to: “Neither appeared changes exceeding normal ranges when the FFP2exhal or community mask was worn.”

Introduction. The reviews you cite, references 7 and 8, are not really comparable. One was specifically looking at exercise and the other more generally. In the case of this study, the Shaw review is the most relevant for recreational exercise. The Kisilinsky review might be more relevant for occupational exercise. It is not clear whether recreational or occupational exercise is the main focus of this paper.

  • It was not our aim to directly compare these two studies. However, these studies show the controversial debate on potential side effects of face masks. We slightly adjusted the text (new text in bold letters Two recent reviews, one scoping review focusing on negative effects by face masks and a systematic review analyzing the effects of face masks during exercise have summarized the current scientific evidence on this topic with different perspectives.

Method. What is the rationale for using PWC130 and 150. Why not measure a fixed power output or % power output are record heart rate? Given the criticism of the Shaw review for focusing on maximal exercise, your choice of a specific submaximal exercise needs expanding on. Appendix D should be put in the method section to provide more detail on PWC130 and 150.

  • We provided more information on why choosing PWC130 and 150 under 2.5 Outcomes and measurements (page 6). In addition, we deleted Appendix D and moved its content to the method section 2.6 data analysis (page 8)

Method. First mention of a warm-up prior to exercise comes at the end of the first physiological responses paragraph. Warm-up should be described in detail earlier on in methods, when talking about the maximal and submaximal tests performed.

  • There was no warm-up period since the tests started with 25 Watt and 50 Watt in trained persons, respectively. However, in the physiological response section we had a formulation which could have been misleading. So, we changed the sentence as follows (new txt in bold letters): “The skin sensor electrode with a temperature of 44° Celsius (C) was placed on the right upper arm over the middle deltoid muscle 10 min prior the start of the ergometer test in order to fulfill the warming requirements of this electrode.” Warming here is only the warming of the electrode which has to have skin contact 10 min in advance in order to produce reliable signals.

Method. Why was respiratory rate calculated from 63-95 seconds of each stage? Those sound like oddly specific numbers.

  • Our aim was to calculate the breathing frequency from the end of each load level. According to our preliminary testings a duration of about 30 seconds would be sufficient to provide a solid data basis (not influence by single extreme values) to calculate berating frequency. However, due to the requirements of a fast furrier transformation 32s had to be used. We provided this information in the text as follows (new text in bold): Breathing frequency was calculated from the end of each load level. However, this interfered with the assessment of perceived exhaustion and perceived respiratory effort, which took place within the last 20 s of each load level. Therefore, and with an additional 5 s buffer, the respiratory rate was calculated from about 30 s before the last 25 s of each load level. Concretely, to calculate the respiratory rate, data from second 63 to second 95 were selected in each 120-second load level. The 512 voltage readings of these 32 seconds were transferred from the time domain to the frequency domain using Fast Fourier Transform (FFT). The input vector of an FFT must have a length of two to the power of n. Thus, in combination with the 16 Hz sampling rate data from 32 s were used.

Method. Were respiratory effort and perceived exertion measured separately, but using the same scale? If so, why? Is there not a more appropriate/specific scale that could be used for respiratory effort?

  • Yes, we used the same scale for assessing respiratory effort and perceived exertion. This scale is commonly used for both during ergometric exercise testing.

Method. It seems strange that the procedures section comes so far into the method. I would have thought this should be one of the first things discussed.

  • We moved the procedure section to an earlier position (2.3 Procedures, page 4).

Results. I don't find the dropout section relevant.

  • We would like to keep this section since reporting dropouts and reasons for dropouts are required according current research reporting standards such as the CONSORT or STROBE Statement.

Results. Some statistical analysis to support the statements regarding differences between groups for age, BMI and physical activity would be useful.

  • We added the statistical analysis for analyzing differences in the three fitness level groups regarding their anthropometrics. Thus, we added the following text to the data analysis section (2.6, page 9): “Differences in anthropometric data between the three fitness level groups were analyzed using a one factorial Analysis of Variance (ANOVA) and Tukey’s honestly significant differences test for post hoc comparison.”
  • In addition, we added PWXmax values of the three fitness level groups (table 1), we somehow have forgotten this in the original manuscript

Table 3. Please highlight any statistically significant differences within the table. The same goes for the figures.

  • We highlighted statistically significant differences in table 3 and the corresponding figures.

Figure 2. The legend for figure 2 is in German. Should at tcpCO2 at PWC130, rather than bei PWC130.

  • we corrected this – thank you.

All captions for tables and figures seem to have the manufacturer of the mask listed as well. This is not necessary as this was given at the first mention of each of the mask.

  • We removed the manufacturer from the figure and table captions.

Round 2

Reviewer 3 Report

The text in lines 514-515, Table 3 and Figure 2a either don't seem to match up or are very confusing.. The text says that PWC130 was significantly lower for FFP2. Table 3 indicates that it is the MedMask condition that was significantly lower. I'm not even sure what Figure 2a indicates, but it is something different from the text and Table 3.

Line 628. I think it would be worth clarifying that there was a significant difference for one condition, that but that this was not classed as relevant. Failing to recognise this does somewhat negate doing that stats tests in the first place.

New Appendix D is not relevant and needlessly adds further length to what is already a long paper and long appendix.

Author Response

Dear reviewer,

thank you again for your feedback.

The text in lines 514-515, Table 3 and Figure 2a: We made a mistake in Table 3 by filling the wrong column. This was corrected. The significant difference occured between FFP2 and no mask for the outcome PWC130, as mentioned in the text and indicated by the bracket with the asterisk in figure 2a.

line 628: We added the following sentence: [...], although PWC130 was slightly and statistically significant reduced when using the FFP2exhal compared to the no mask condition.

New Appendix D:

Here we have a conflict between you and another reviewer. Another reviewer needed more information about the time intervals between the experimental conditions. We understand that the manuscript should not become even longer, but we think that there may well be readers, such as the other reviewer, who can then find exactly such sepcific information in the appendix. We therefore would like to keep Appendix D
